# Serum Neurofilament Light Chain and Glial Fibrillary Acidic Protein as Potential Diagnostic Biomarkers in Autism Spectrum Disorders: A Preliminary Study

**DOI:** 10.3390/ijms24033057

**Published:** 2023-02-03

**Authors:** Marta Simone, Andrea De Giacomo, Roberto Palumbi, Claudia Palazzo, Giuseppe Lucisano, Francesco Pompamea, Stefania Micella, Mara Pascali, Alessandra Gabellone, Lucia Marzulli, Paola Giordano, Concetta Domenica Gargano, Lucia Margari, Antonio Frigeri, Maddalena Ruggieri

**Affiliations:** 1Regenerative and Precision Medicine Department and Jonic Area (DiMePRe-J), University of Bari “Aldo Moro”, 70124 Bari, Italy; 2Translational Biomedicine and Neuroscience Department (DiBraiN), University of Bari “Aldo Moro”, 70124 Bari, Italy; 3Interdisciplinary Department of Medicine, University of Bari “Aldo Moro”, 70124 Bari, Italy

**Keywords:** autism spectrum disorder, glial fibrillary acidic protein, neurobiology, neuroinflammation, glial fibrillary acidic protein, neurofilament, neuroimmune

## Abstract

Autism spectrum disorder (ASD) is one of the most common neurodevelopment disorders, characterized by a multifactorial etiology based on the interaction of genetic and environmental factors. Recent evidence supports the neurobiological hypothesis based on neuroinflammation theory. To date, there are no sufficiently validated diagnostic and prognostic biomarkers for ASD. Therefore, we decided to investigate the potential diagnostic role for ASD of two biomarkers well known for other neurological inflammatory conditions: the glial fibrillary acidic protein (GFAP) and the neurofilament (Nfl). Nfl and GFAP serum levels were analyzed using SiMoA technology in a group of ASD patients and in a healthy control group (CTRS), age- and gender-matched. Then we investigated the distribution, frequency, and correlation between serum Nfl and GFAP levels and clinical data among the ASD group. The comparison of Nfl and GFAP serum levels between ASD children and the control group showed a mean value of these two markers significantly higher in the ASD group (sNfL mean value ASD pt 6.86 pg/mL median value ASD pt 5.7 pg/mL; mean value CTRS 3.55 pg/mL; median value CTRS 3.1 pg; GFAP mean value ASD pt 205.7 pg/mL median value ASD pt 155.4 pg/mL; mean value CTRS 77.12 pg/mL; median value CTRS 63.94 pg/mL). Interestingly, we also found a statistically significant positive correlation between GFAP levels and hyperactivity symptoms (*p*-value <0.001). Further investigations using larger groups are necessary to confirm our data and to verify in more depth the potential correlation between these biomarkers and ASD clinical features, such as the severity of the core symptoms, the presence of associated symptoms, and/or the evaluation of a therapeutic intervention. However, these data not only might shed a light on the neurobiology of ASD, supporting the neuroinflammation and neurodegeneration hypothesis, but they also might support the use of these biomarkers in the early diagnosis of ASD, to longitudinally monitor the disease activity, and even more as future prognostic biomarkers.

## 1. Introduction

Autism spectrum disorder (ASD) is a neurodevelopmental disorder characterized by impairment in different brain functional areas such as communication, social interaction, and restricted behaviors [1]. The diagnosis is often made as early as 18 months of age, but most patients are not formally diagnosed until 5 years old [2]. Many comorbidities, such as intellectual disability, epilepsy, Tourette’s syndrome, difficulty sleeping, and gastrointestinal dysfunctions, have been described [2]. Moreover, most people with ASD have sensory processing disorder (SPD), which is an inability to respond behaviorally to sensory input, experienced due to impaired sensory input such as sound, touch, body movement, sight, taste, and smell, with the prevalence of SPD in autism estimated to be around 90%. Thus, it can be very challenging for parents when traveling with children with autism due to reactions to sensory stimuli resulting in behavioral problems, e.g., crying, screaming, shouting, kicking, banging toys, and throwing, which lead to self-injury and danger to themselves and others, including the driver. Afif I.Y. and colleagues have investigated the physiological effect of deep pressure created by the hug machine portable seat (AHMPS) in children with autism spectrum disorders (ASD) on public transportation settings [3,4]. In both studies, they found an improvement in anxiety symptoms and neurobiological stress parameters when children wore this portable device.

The prevalence of ASD is highly variable across the world: it is estimated to be 2.5% in USA, 1.5 % in Denmark, Finland, and Sweden, and 1.7% in Great Britain [5]. According to a recent report, in Italy, 1 child in 77 (aged 9 to 7 years) is diagnosed with autism [6]. Autism is defined as a disorder characterized by a multifactorial etiology, with strong neurogenetic and environmental factor interaction [7,8,9,10,11]. However, since it is unclear which is the most likely etiopathogenetic mechanism, there are currently no specific laboratory diagnostic tests [12]. Strong genetic association with ASD is supported by data from monozygotic twins; high concordance and large-scale genetic studies have revealed many autism risk factors, each with relatively low penetrance [12,13].

Usually, environmental factors associated with ASD occur early in gestation, during the first or second trimester [14,15]. For example, prenatal exposure to valproic acid, an anticonvulsant drug during the first trimester of gestation, significantly increases the incidence of autism in children [15]. In addition, other maternal and prenatal risk factors are associated with a high risk of ASD (e.g., maternal diabetes, exposure to environmental pollutants, hormonal factors) [15]. Approximately 5% of individuals exposed in utero to thalidomide between day 24 and 36 of gestation and misoprostol developed ASD [16]. Several studies have suggested that maternal infections associated with fever, immune activation, significant bleeding during the second trimester, or occurrences of cytomegalovirus infection during the third trimesters are risk factors for ASD [17,18,19]. Other factors have been linked with increased risk of ASD, including older paternal age, prenatal stress, maternal diabetes, and obesity [16].

In addition to these findings, other studies reported an association between ASD and the impairment in specific brain regions’ development, such as the amygdala, the cerebellum, and other regions of the brain [20,21,22,23]. Moreover, it has been demonstrated that the brains of children with ASD tend to grow faster than usual just after birth, followed by normal or relatively slower growth during childhood; furthermore, this atypical brain growth in ASD seems to involve specific regions [24].

Nevertheless, although extensive studies have been carried out on autism, there are still several controversies about its neurobiological mechanisms. Evidence of neurodegeneration and neuroinflammation has been found in cases of children with ASD who experienced progressive loss of neurological function; these findings were associated with abnormal microglial cells and astrocyte activation, evidence of oxidative stress, and with the presence of proinflammatory cytokines and neuronal cell loss [25,26,27,28]. Neuroinflammation is a complex mechanism that involves the activation of different types of cells, such as astrocytes and microglia cells, in order to repair the damage [18,29]. Glial fibrillary acidic protein (GFAP) is the hallmark intermediate filament protein of the astrocytes, and it is a marker of astroglial activation. High levels of this protein have been found in the cerebrospinal fluid of children with autism, Rett’s syndrome [30], and in many neurodegenerative diseases [31]. Moreover, recent studies supported the promising role of the blood GFAP level as a diagnostic and prognostic biomarker of brain and spinal cord disorders [32,33].

More recently it has been shown that serum GFAP levels may be associated with the severity of ASD [34]. Interestingly, a recent study found that the number of astrocytes decreased in all prefrontal areas, both in gray and white matter, as well as an increased abnormal activation state of GFAP+ astrocytes [35]. Overall, these findings are consistent with astrocyte disruption in ASD. Neuroaxonal damage is the pathological substrate of permanent disability in many neurological disorders [36]. The neurofilament protein levels rise upon neuroaxonal damage in the cerebrospinal fluid (CSF) and/or in the bloodstream [35]. The light subunit of neurofilament protein (Nfl) is a sensitive indicator of neuronal injury and neurodegeneration [37]. Its levels are elevated in cerebrospinal fluid (CSF) and/or blood proportionally to the degree of axonal damage in a variety of neurological disorders, including inflammatory [38], cognitive decline and neurodegenerative [39], traumatic [40], and cerebrovascular diseases [41]. It is a CSF-sensitive and specific marker for white matter axonal injury in patients with multiple sclerosis showing white matter changes during follow-up studies [42]. Interestingly, white matter abnormalities have been reported in ASD, and these alterations have been associated with abnormalities in the functional connectivity [43]. Karahanoğlu et al. (2018) also concluded that ASD was characterized by altered white matter development from childhood to early adulthood, and those changes might underlie abnormal brain function and contribute to core features [44]. To date, besides a case-control study investigating serum Nfl (sNfl) in autistic children [45], the evaluation of both Nfl and GFAP serum levels in ASD patients has never been investigated.

Early diagnosis of autism leads to targeted rehabilitation interventions that can improve the natural history of the disease. Identifying diagnostic and possibly prognostic biomarkers at an early stage will help the clinician in choosing a targeted intervention for that type of patient.

The novelty of our study lies in the simultaneous analysis of two serum biomarkers indicating processes of neuroinflammation and neurodegeneration using the SiMoA technology.

Thus, the main objective of this study was the evaluation of these biomarkers to find possible diagnostic value and correlation with disease severity. The first aim was to assess Nfl and GFAP serum levels in a population of pediatric patients divided into two groups: the first group diagnosed with ASD and the second with neurotypical development (control group). The second aim was to investigate, among ASD, the relationship between serum NfL and GFAP levels and the clinical features obtained by neuropsychiatric assessment.

## 2. Results

A total of 71 children were recruited: 42 patients with ASD diagnosis were included in the clinical sample and 29 age- and gender-matched neurotypical development subjects (CTRS) were selected for the control group among patients admitted to the pediatric department for minor issues.

Sociodemographic and clinical data of ASD patients are shown in Table 1: The ASD sample included 33 males and 9 females (male 78.57 %; female 21.43%), between 2 and 16 years old (mean value: 6.98 years; median 6 years). The control group included 23 males and 6 females (male 79.31%; female 20.69%), between 3 and 16 years old (mean value 7.21; median 7 years). In the ASD group, an intellectual disability was diagnosed in 21 patients (50%), and a language impairment was found in 29 patients (69.05%). Social impairment and restricted/repetitive behaviors were evaluated using Autism Diagnostic Observation Schedule 2 (ADOS-2) [46]: 29 patients (69.05%) showed a total score between 8–16 points, corresponding to mild-moderate ASD, and 13 patients (30.95%) showed a total score over 16 points, corresponding to severe ASD.

Moreover, the following comorbidities were identified: 7 patients (16.67%) had a diagnosis of Attention Deficit Hyperactivity Disorder (ADHD), 17 patients (40.48%) manifested hyperactivity/impulsivity symptoms under the threshold for ADHD, and 1 patient (2.38%) had a diagnosis of a learning disorder. The instrumental diagnostic assessment evidenced the presence of nonspecific electroencephalogram (EEG) alterations in 4 patients (9.52%), and only 2 patients (4.76%) presented Magnetic Resonance Imaging (MRI) alterations; both findings were without clinical significance.

The comparison of Nfl and GFAP serum levels between ASD children and control group is shown in Figure 1. The results evidenced a mean value of the serum levels of Nfl and GFAP significantly higher (*p* < 0.0001) in the ASD sample compared to controls (sNfl mean value ASD pt 6.86 pg/mL median value ASD pt 5.7 pg/mL; mean value CTRS 3.55 pg/mL; median value CTRS 3.1 pg; GFAP mean value ASD pt 205,7 pg/mL median value ASD pt 155.4 pg/mL; mean value CTRS 77.12 pg/mL; median value CTRS 63.94 pg/mL).

Then, in the ASD sample, Nfl and GFAP serum levels were compared between patients characterized by different neuropsychiatric and medical features. Although these results did not show statistical significance, Nfl and GFAP serum levels were higher in ASD subjects showing EEG and MRI abnormalities, speech impairment, intellectual disability, and higher ADOS-2 scores. Interestingly, GFAP serum levels were significantly higher in patients with hyperactivity (*p* value < 0.001). The matrix Spearman’s correlation test is shown in Table 2; a positive trend of significance was found when GFAP serum levels were correlated with the symptom hyperactivity. Moreover, we found another positive trend of significance between both serum biomarker levels.

In the “Appendix A” are collected figures of the star plots showing the results of the correlation test of sGFAP and sNfl levels (Appendix A).

## 3. Discussion

Autism Spectrum Disorder is one of the most common neurodevelopment disorders, and the research of reliable diagnostic and/or prognostic biomarkers offers new challenges. The key results of this detailed analysis of serum Nfl and GFAP measured by Simoa in a well-characterized cohort of ASD and control group patients are (1) a mean value of Nfl and GFAP serum levels that is significantly higher in the ASD sample compared to controls; 2) a significant positive correlation between GFAP serum levels and hyperactivity. The Simoa technique allows one to measure very low concentrations of sNFl and sGFAP; it has previously been used by Wei-chao-He [45], but to our knowledge, our investigation represents the first evaluation of both Nfl and GFAP serum levels between two groups using this ultrasensitive technique.

Serum levels of Nfl and GFAP have been mostly investigated in some neurological conditions rather than in ASD or other neurodevelopmental disorders. Firstly, serum GFAP levels were studied as potentially informative for the prognostic and diagnostic evaluation of some neoplastic forms, such as gliomas [47]. Foerch et al. reported how the variation in sGFAP levels evaluated at 4–5 h from a cerebrovascular event allows for the recognition of the presence of an intracerebral hemorrhage from an ischemic event [48]. Fenerberg et al. reported the validity of GFAP as a potential biomarker in some forms of narcolepsy [49]. Papa et al. reported the possibility of identifying the degradation products of GFAP up to one hour after a traumatic event in the brain, and this finding is also associated with the severity of the damage [50]. More recently, two studies supported the clinical use of the sGFAP in monitoring traumatic brain injury and other neurological conditions, as neurodegenerative or neuroinflammatory diseases [32,33].

As for GFAP, the prognostic and diagnostic role of serum Nfl has been investigated: elevated levels were identified in many pathologies such as Huntington’s chorea, spinal cord lesions, neurodegenerative diseases such as multiple sclerosis, Parkinson’s and Alzheimer’s, and various forms of dementia [51].

In ASD, the data about these biomarkers’ diagnostic and prognostic value are very limited. However, our findings are coherent with previous studies. Some authors investigated the correlation of high serum Nfl values and severity of ASD symptoms, comparing serum Nfl levels in a group of ASD patients (age 3–8 years) to a control group of peers [45]. Samples were analyzed using the SiMoA technique. The results showed a statistically significant difference in serum Nfl values between the two groups (*p* < 0.0001). To assess the correlation between serum Nfl and the severity of ASD symptoms, serum values were compared to the patient score obtained through the structured CARS interview (which explores the severity of ASD symptoms), highlighting a positive correlation between the increase in serum NFl values and high CARS scores. Moreover, through an analysis of postmortem histological samples, several studies have revealed an aberrant astrocytic activity in different regions of the brain, highlighting a high astrocytic infiltration by immunohistochemical assays with an intense detection of GFAP. In 2015, Crawford et al. compared the levels of astrocyte activation markers in postmortem histological specimens between ASD patients and normal development subjects [52]. The results showed a statistically significant increase in GFAP values in the white matter of the anterior cingulate cortex, the Broca region, concluding that there was an alteration of the connections in the white matter of this region as a possible pathogenetic mechanism of ASD. Menassa et al. showed an increase in astrocytic activity witnessed by an increase in GFAP levels, at the level of the primary olfactory cortex, hypothesizing the role of astrocytic activity in the hypo/hyper sensory phenomena of ASD [53]. In a very recent study, the authors investigated the expression of immunity-related genes in postmortem brain tissues from patients affected by six main brain disorders, including ASD [54]. They found that the expression of several immunity genes significantly differed in autistic patients from other neuropsychiatric conditions, highlighting a sex-difference expression as well. The most common variation of transcriptomes involved genes associated with the microglia activation, a process that might lead to a disruption of neuronal maturation and synaptogenesis. Evidence of increased GFAP+ astrocyte activation was also found in all prefrontal cortexes of ASD patients’ postmortem samples [35]. In 2016, Cetin et al. conducted a study with the aim of evaluating the serum levels of GFAP in ASD patients: the study involved a cohort of 49 children divided into a group of 22 ASD patients and 27 controls [55]. Measurements of GFAP serum levels were carried out by immunoenzymatic method (ELISA). The results showed significantly low values of GFAP in the ASD sample compared to the control sample, assuming that the products resulting from the excessive activation of astrocytes were not released into the circulation, tending to precipitate within the reactive astrocytes. In another study, Wang et al. in 2017 compared the sGFAP levels in ASD children (3 to 8 years) and in a control group [34]. The results showed a statistically significant difference in GFAP values in the ASD group if compared to controls. Furthermore, a positive correlation was demonstrated between GFAP levels and CARS scores. Another significant result of our study is that we found a stronger positive correlation between sGFAP and “hyperactivity” in autistic patients. To our knowledge, we did not find previous studies on the relationship between this symptom or attention deficit/hyperactivity disorder and sGFAP. However, since GFAP is mostly recognized as a biomarker of neuroinflammation, we could assume that in autistic patients who present a more severe functional profile and who also present clinically significant hyperactivity traits or other comorbidities, the mechanisms of neuroinflammation might be more crucial than the neurodegeneration processes in determining the severity of the clinical phenotype.

A recent review reported evidence that supports the alteration of innate immunity in ASD, collecting data about aberrant immunity cell function and evidence of neuroinflammation and microglia activation [56]. Another review found that an early and uncontrolled brain immune activation in ASD, mediated by prenatal infections or other immunogenic stimuli, might lead to an abnormal neurogenesis and synaptogenesis, mediated mostly by astrocytic activation in specific brain regions such as the prefrontal, temporal, cerebellar, and anterior cingulate cortex regions [57]. As supported by a recent study, astrocytic and microglial activation is promoted by proinflammatory cytokines, such as IL1, IL6, IL8, IFN, and TNK-alpha, which consolidates and maintain a continuous brain immune dysregulation and oxidative stress, which might affect neurodevelopment pathways [58].

Taken together, these data strongly support the current hypothesis that neuroinflammation might play a potential role in the neurobiology of autism, mostly through the activation of glial cells such as astrocytes and microglia [18,20,27,28,29,54,55,56,57,58].

## 4. Materials and Methods

### 4.1. Study Samples

The participants were recruited at the Child Neuropsychiatric Unit, University/Hospital Policlinico of Bari, between July 2021 and April 2022. The sample selection was based on patients, who referred to our unit as a first evaluation or for follow-up, diagnosed with ASD, according to the Diagnostic and Statistical Manual of Mental Disorders—Fifth Edition [1] Criteria by a trained child neuropsychiatrist. All the ASD patients were treatment-naïve.

A sex and age-matched control group was recruited between typical neurodevelopment patients referred to General Pediatric Unit for minor illnesses, excluding autoimmune, inflammatory, infective, and major neurological diseases.

All the participants underwent blood sampling, and sera were tested for Nfl and GFAP serum levels. Demographic (age and gender), clinical, brain magnetic resonance imaging (MRI), electroencephalogram (EEG), and neuropsychiatric data were recorded for all the patients. Figure 2 explains the workflow of the assessment of this study.

For all the participants, written informed consent was collected by the caregivers and the study was approved by the Local Ethics Committee of the University Hospital—Policlinico of Bari (Approval Number: 0005379|21/01/2021).

### 4.2. Serum Nfl and GFAP Assessment

Nfl and GFAP serum levels were analyzed by SiMoA technology using the Quanterix-SRX instrument. Nfl concentrations were measured via the commercially available SiMoA NF-light Advantage Kit (Quanterix, Billerica, MA, USA). Serum GFAP levels were measured via the SiMoA GFAP Discovery kit (Quanterix, Billerica, MA, USA) following the manufacturer’s instructions. Briefly, samples were tested blindly and in duplicate, and two quality controls (high-concentration and low-concentration quality control) were run, in duplicate as well, on each plate for each run necessary to complete the study. Samples were run with a 4-fold dilution and results were compensated for this dilution. Nfl and GFAP concentrations (pg/mL) were calculated using a standard curve made from a sample of known concentrations in triplicate, as recommended by the manufacturer. For each protein, intra-assay coefficient of variation (CV) values were calculated by the SRX-analyzer software from technical replicate measures of specimens assayed within a single run. CV for these samples was <1%.

### 4.3. Neuropsychiatric Assessment

All the ASD participants underwent the following neuropsychiatric evaluation: Intelligence quotient (IQ) was evaluated for all ASD patients. Patients with speech impairment underwent a non-verbal IQ evaluation using the Leiter-R Visual and Reasoning battery [59]. This test is composed of 10 subtests that evaluate spatial visualization and reasoning abilities. Patients without speech problems underwent IQ evaluation through Wechler Scales: Wechsler Preschool and Primary Scale of Intelligence—Third Edition (WPPSI-III) [60] was used for patients from 2 yrs to 7 yrs obtaining total IQ score, verbal IQ score, performance capacities, processing abilities, and general language; Wechsler Intelligence Scale for Children, Fourth Edition (WISC-IV) [61] was used in patients from 6 yrs to 16 yrs to evaluate total IQ score, verbal capacities, nonverbal capacities, memory, and elaboration speed. In this study, we considered only the total IQ score.

Autism Diagnostic Observation Schedule-2 (ADOS-2) [46] is a semi-structured clinical observation used to evaluate symptoms of ASD. It has five different modules used for different ages to evaluate “Social Interaction” and “Stereotype behaviors and restricted interests”. Each section is composed of different items and each of them has a score from 0 to 2 according to the severity of specific behaviors. Each module has a diagnostic algorithm in which a score is used to evaluate the “Social Affect” dimension and “Restricted and Repetitive Behaviors”. The total score can be used to determine the severity of ASD symptoms: mild (level 1), moderate (level 2), and severe (level 3).

The language skills evaluation was conducted using the Language Evaluation Test (Test di Valutazione del Linguaggio, Test TVL) [62], a standardized protocol used to evaluate language development. This test is used to assess both receptive and expressive language abilities, including phonetic and syntactic aspects of expressive language.

For ASD patients who had IQ >70 and who already started primary school, writing, reading, and maths abilities were evaluated using the following standardized protocols: MT-3 test (Italian version) for reading skills [63], AC-MT 3 Test (Italian version) [64] for maths abilities and the battery “Dislessia e Disortografia Evolutiva 2 (DDE-2) [65] for the writing skills assessment. For ASD patients with cognitive impairment, a nonstandardized evaluation of scholastic abilities was conducted. Attention skills and hyperactivity symptoms were assessed using the Conners’ Parent Rating Scales [66].

### 4.4. Statistical Analysis

Sociodemographic data, including age and sex; clinical data, including IQ, language impairment, the severity of ASD symptoms, and total ADOS-2 score; and medical data, including EEG abnormalities, MRI abnormalities, and blood serum levels of Nfl and GFAP, were collected in structured databases exclusive for this research. Qualitative data were expressed in frequencies and quantitative data in means and medians. Statistical analysis was performed using IBM SPSS. Initially, we compared serum Nfl and GFAP levels of ASD patients and the neurotypical control group using the Mann–Whitney test. Then, we evaluated the distribution, frequency, and correlation between serum Nfl and GFAP levels and clinical data among the ASD group patients using the Mann–Whitney test and nonparametric Spearman correlation test.

## 5. Conclusions

In conclusion, to our knowledge, the present study is the first one investigating both serum GFAP and Nfl in autistic children as potential diagnostic biomarkers. Indeed, we found that both of these markers are significantly higher in ASD patients when compared to healthy controls. Moreover, we found a positive significant correlation between sGFAP and the hyperactivity symptom. These preliminary data require further investigations to verify their potential diagnostic and prognostic purposes. Future studies might support the use of these biomarkers in the early diagnosis of ASD or in monitoring the progress of the disease, and eventually the treatment response.

### Limitations

The main weakness of this preliminary study is the small size of the ASD group. Further investigations using larger groups are necessary to confirm our data and to verify in more depth the potential correlation between these biomarkers and ASD clinical features, such as the severity of the core symptoms, the presence of associated symptoms, and/or the evaluation of a therapeutic intervention. Nevertheless, even if this study is a preliminary investigation, the presence of significantly high levels of both serum biomarker values in the ASD group might support the hypothesis of mechanisms of neuroinflammation and neurodegeneration in the neurobiology of autism, and this evidence might have a higher impact on further studies.

## Figures and Tables

**Figure 1 ijms-24-03057-f001:**
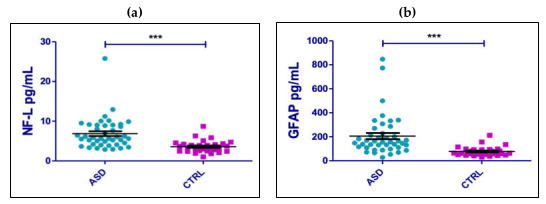
Comparisons of sNfl (**a**) and sGFAP (**b**) values between ASD patients and control group. Mann Whitney test *** *p* value < 0.001.

**Figure 2 ijms-24-03057-f002:**
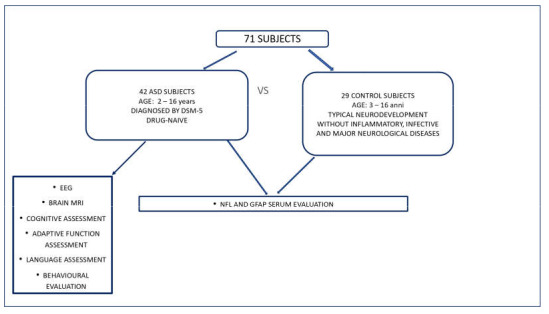
Workflow of the assessment of this study.

**Table 1 ijms-24-03057-t001:** Sociodemographic and clinical features of ASD patients.

Number	42
**Age (yrs)** (mean value; median)	6.98; 6
**Gender** Male Female	33 9
**Level of Severity ASD (%)** **1** **2** **3**	42.86% 38.1% 19.05%
**Frequency of EEG abnormalities (%)**	9.52%
**Frequency of MRI alterations (%)**	4.76%
**Learning disorder (%)**	2.38%
**ADHD (%)**	16.67%
**Hyperactivity (%)**	40.48%
**Language impairment (%)**	69.05%
**Intellectual disability (%)**	50%
**ADOS-2 score (%)** **8–11** **11–16** **>16**	23.81% 45.24% 30.95%

**Table 2 ijms-24-03057-t002:** Matrix of Spearman’s correlation test results.

	Age	Gender	ASD Level	EEG Abnormalities	MRI Abnormalities	Learning Disorder	ADHD	Hyperactivity	Language Impairment	Intellectual Disability	Nfl	GFAP	Total Scores ADOS-2
**Age**	1												
**Gender**	−0.26	1											
**ASD level**	−0.09	−0.15	1										
**EEG Abnormalities**	−0.19	−0.03	−0.1	1									
**MRI Abnormalities**	0.01	−0.16	−0.24	−0.07	1								
**Learning Disorder**	0.18	0.08	−0.17	−0.05	−0.03	1							
**ADHD**	−0.05	0.08	0.11	−0.15	−0.1	−0.07	1						
**Hyperactivity**	−0.08	−0.04	0.28	−0.1	0.04	−0.13	**0.37**	1					
**Language impairment**	**−0.4**	0.03	**0.5**	0.04	−0.09	−0.23	0.02	0.24	1				
**Intellectual Disability**	−0.13	−0.06	**0.44**	0	−0.22	−0.16	−0.06	0.24	**0.36**	1			
**sNfl**	**−0.39**	0.05	0.17	0.04	−0.13	−0.06	0.14	0.17	0.21	0.21	1		
**sGFAP**	**−0.72**	0.22	0.14	0.06	0.03	−0.23	−0.04	**0.31**	0.22	0.26	**0.56**	1	
**Total Scores ADOS-2**	−0.22	−0.15	**0.63**	−0.04	−0.15	−0.1	0.12	0.08	**0.34**	0.26	0.11	0.21	1

## Data Availability

Not applicable.

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
