# Peer review of "Serum Neurofilament Light Chain and Glial Fibrillary Acidic Protein as Potential Diagnostic Biomarkers in Autism Spectrum Disorders: A Preliminary Study"

_ijms, 2023, doi:10.3390/ijms24033057_

Round 1

Reviewer 1 Report

The authors present their evaluation of novel biomarkers of ASD, the manuscript is well written however the small sample size limits its strength. In line number 98 "we evaluated these biomarkers to find their prognostic value in early diagnosis 98 of the disease", how was the prognostic value assessed, this should be included.

Author Response

Point 1: The authors present their evaluation of novel biomarkers of ASD, the manuscript is well written however the small sample size limits its strength. In line number 98 "we evaluated these biomarkers to find their prognostic value in early diagnosis 98 of the disease", how was the prognostic value assessed, this should be included.

Response 1: Thank you for your comment and appreciation. We are aware of the limitations of the study, but we are planning to continue it to increase the sample size. Now we have modified line 98: ”we evaluated these biomarkers to find possible diagnostic value and correlation with disease severity”.

Reviewer 2 Report

1.      The abstract requires the addition of quantitative results.

2.      Please add the abstract's "take-home" message, the current form was insufficient.

3.      Put the keywords in a new order based on alphabetical order.

4.      The Reviewer do not see the novel in the present article. My examination revealed that several similar previous publications appear to appropriately address the issues you have brought up in the current submission. Please emphasize it more advance in the introduction section if there are any more truly something really new.

5.      The work, novelty, and limitations of similar prior studies must be explained in the introduction section to highlight the research gaps that the current study aims to fill.

6.      The last paragraph of the introduction section should be the objective of the present study.

7.      Since the present study discuss about autism spectrum disorder, the authors need to explain about anxiety in children with autism spectrum disorder. It is a vital topic that authors must provide in the introduction and/or discussion section. Additionally, the MDPI's suggested reverence should be taken to substantiate this explanation as follows: Physiological Effect of Deep Pressure in Reducing Anxiety of Children with ASD during Traveling: A Public Transportation Setting. Bioengineering 2022, 9, 157. https://doi.org/10.3390/bioengineering9040157

8.      To enhance the understandability of the section on materials and methods easier for them to understand rather than just depending on the main text as it exists at the moment, the authors could add additional illustrations in the form of figures that explain the workflow of the present study.

9.      What is the baseline of sample selection? Is there any protocol, standard, or basis that has been followed? It is unclear since the patient is very heterogeneous with a small number. The resonance involved impacts the present result makes this study flaws. One major reason for rejecting this paper.

10.   It is required to include additional information on tools, such as the manufacturer, the country, and the specification.

11.   Valuable information that must be included in the publication refers to the inaccuracy and intolerance of the experimental setup used in this inquiry.

12.   Outcomes must be compared to similar past research.

13.   The authors need to improve the discussion in the present article become more comprehensive. The present form was insufficient.

14.   Line 294-301, the reviewer believes there is other limitation that not mentioned.

15.   Line 303-306, please develop the conclusion. It is not comprehensive.

16.   In the conclusion, please explain the further research.

17.   The reference needs to be enriched from the literature published five years back. MDPI reference is strongly recommended.

18.   The authors occasionally created paragraphs in the entire document that were just one or two phrases long, which made the explanation difficult to understand. To make their explanation into a longer, more thorough paragraph, the authors should expand it. It is advised to use at least three sentences in a paragraph, with one serving as the primary sentence and the others as supporting phrases.

19.   The manuscript needs to be proofread by the authors since it has grammatical and language issues.

20.   It is suggested to the authors for providing graphical abstract in the system after revision.

Author Response

Point 1: The abstract requires the addition of quantitative results.

Response 1: Thank you for your suggestion. Now we have added the quantitative results in the abstract.

Point 2: Please add the abstract's "take-home" message, the current form was insufficient.

Response 2: Thank you. We have modified now the conclusion of the abstract

Point 3: Put the keywords in a new order based on alphabetical order.

Response 3: Thank you. We placed the keywords in alphabetical order.

Point 4: The Reviewer do not see the novel in the present article. My examination revealed that several similar previous publications appear to appropriately address the issues you have brought up in the current submission. Please emphasize it more advance in the introduction section if there are any more truly something really new.

Response 4: Thank you for this suggestion. Now we added new elements which could describe and emphasize better the novelty of our study.

Point 5: The work, novelty, and limitations of similar prior studies must be explained in the introduction section to highlight the research gaps that the current study aims to fill.

Response 5: Thank your comment. Now we added in the “Introduction” section new elements that highlight the novelty of our study aiming to fill current research gaps.

Point 6: The last paragraph of the introduction section should be the objective of the present study.

Response 6: Thank you. Now we have modified the last paragraph of the introduction.

Point 7: Since the present study discuss about autism spectrum disorder, the authors need to explain about anxiety in children with autism spectrum disorder. It is a vital topic that authors must provide in the introduction and/or discussion section. Additionally, the MDPI's suggested reverence should be taken to substantiate this explanation as follows: Physiological Effect of Deep Pressure in Reducing Anxiety of Children with ASD during Traveling: A Public Transportation Setting. Bioengineering 2022, 9, 157. https://doi.org/10.3390/bioengineering9040157

Response 7: Thank you for this suggestion. Now we have added a paragraph in the introduction section explaining the sensorial problem in ASD children, mentioning this citation.

Point 8: To enhance the understandability of the section on materials and methods easier for them to understand rather than just depending on the main text as it exists at the moment, the authors could add additional illustrations in the form of figures that explain the workflow of the present study.

Response 8: Thank you for your comment. Now we added a workflow illustration (Figure 1) to describe the assessment of our study better.

Point 9: What is the baseline of sample selection? Is there any protocol, standard, or basis that has been followed? It is unclear since the patient is very heterogeneous with a small number. The resonance involved impacts the present result makes this study flaws. One major reason for rejecting this paper.

Response 9: We appreciate a lot this comment. Now we have tried to describe better the sample selection for both patients and control in the Material and Methods section: “Study Sample”

Point 10: It is required to include additional information on tools, such as the manufacturer, the country, and the specification.

Response 10: Thank you for this comment, but you can find some information about “SiMoa technology” in the 2.2 paragraph section “ Serum NfL and GFAP assessment”. Please do not hesitate to contact us if other details are necessary.

Point 11: Valuable information that must be included in the publication refers to the inaccuracy and intolerance of the experimental setup used in this inquiry.

Response 11: Thank you for your suggestion. Our patients underwent routine non-invasive tests, safe and well tolerated by children.

Point 12: Outcomes must be compared to similar past research.

Response 12: thank you. We added other references about similar research on the same findings.

Point 13: The authors need to improve the discussion in the present article become more comprehensive. The present form was insufficient.

Response 13: thank you for this suggestion. As requested, we edited the discussion and added more comprehensive references, including recent reviews on the topic.

Point 14: Line 294-301, the reviewer believes there is other limitation that not mentioned.

Response 14: Thank you for the comments. In the “Limitation” section we included all the limitations of this study.

Point 15: Line 303-306, please develop the conclusion. It is not comprehensive.

Response 15: thank you. We better developed the conclusions of the manuscript.

Point 16: In the conclusion, please explain the further research.

Response 16: as written in the previous response, in the conclusions, we better explained the main aims of future investigations.

Point 17: The reference needs to be enriched from the literature published five years back. MDPI reference is strongly recommended.

Response 17: thank you. We added more recent references in the manuscript.

Point 18: The authors occasionally created paragraphs in the entire document that were just one or two phrases long, which made the explanation difficult to understand. To make their explanation into a longer, more thorough paragraph, the authors should expand it. It is advised to use at least three sentences in a paragraph, with one serving as the primary sentence and the others as supporting phrases.

Response 18: Thank you for the valuable comments. When necessary, we expanded one paragraph including longer explanations as suggested.

Point 19: The manuscript needs to be proofread by the authors since it has grammatical and language issues.

Response 19: Thank you for the suggestions. The grammar and language have been checked by an English native speaker.

Point 20: It is suggested to the authors for providing graphical abstract in the system after revision.

Response 20: Thanks for the useful suggestion. In the current study, unfortunately, we do not have sufficient elements to provide a graphical abstract. In future studies, we intend to expand the actual results and eventually create a graphical abstract.

Reviewer 3 Report

This manuscript describes that serum GFAP and NfL subunit levels are significantly elevated in peripheral blood of ASD children, which are measured by SiMoA technology. Neurodevelopmental disorders including ASD and ADHD are often difficult to be diagnosed or to be distinguished between the NDDs. Identification of biological markers of NDDs as well as many other psychiatric disorders in peripherally obtainable samples may be critical and challenging targets. In this sense, the present study gives possible hope to clinicians, although the data are not very novel.

The data are not very comprehensive as the authors noted that this is preliminary study. The authors should clearly describe how the present data in the manuscript are different from (or the same as) some previous papers showing the similar results. In particular, a previous work by Wang, J (appeared in Int. J. Dev. Neurosci. 2017) seems to demonstrate almost the same results.

SiMoA technology should be mentioned in Introduction; the principals underlying the method, the merit of the technology (compared to other conventional methods such as ELISA), the reason why the authors employed it, .

There is some inconsistency in writing such as NFL, Nfl, or NfL. 

Author Response

Response to Reviewer 3 Comments

Point 1: The data are not very comprehensive as the authors noted that this is preliminary study. The authors should clearly describe how the present data in the manuscript are different from (or the same as) some previous papers showing the similar results. In particular, a previous work by Wang, J (appeared in Int. J. Dev. Neurosci. 2017) seems to demonstrate almost the same results.

Response 1: Thank you for the comments. Indeed, this study reports just preliminary results. Anyway, in several sections of the manuscript (last sentences of the “Introduction” and in the first paragraph of the discussion), we specified that, to our knowledge, this is the first study investigating both sGFAP and sNfl in ASD patients using SiMoA technology. As correctly reported by the reviewer, Wang et al. investigated only the sNfl. Moreover, as described in the “Results” and “Discussion”, we found that sGFAP levels were positively correlated with the symptom hyperactivity in ASD patients. This is another result that, to our knowledge, has never been reported so far.

Point 2: SiMoA technology should be mentioned in Introduction; the principals underlying the method, the merit of the technology (compared to other conventional methods such as ELISA), the reason why the authors employed it.

Response 2: Thanks for the suggestion. We added in the introduction the merit of the technology used.

Point 3: There is some inconsistency in writing such as NFL, Nfl, or NfL.

Response 3: Thanks for the comments. We edited the manuscript using the correct acronym Nfl through the whole text.

Round 2

Reviewer 2 Report

I have other comments as the response to authors revised version. In the limitations section, the authors explain about the small size of ASD group. The authors would refer ASD study using small group, and explain the present study as preliminary results that would be higher impact from following study in the further. Suggested relevant reference as follows: Effect of Short-Term Deep-Pressure Portable Seat on Behavioral and Biological Stress in Children with Autism Spectrum Disorders: A Pilot Study. Bioengineering 2022, 9, 48. https://doi.org/10.3390/bioengineering9020048

Author Response

Point 1: I have other comments as the response to authors revised version. In the limitations section, the authors explain about the small size of ASD group. The authors would refer ASD study using small group, and explain the present study as preliminary results that would be higher impact from following study in the further. Suggested relevant reference as follows: Effect of Short-Term Deep-Pressure Portable Seat on Behavioral and Biological Stress in Children with Autism Spectrum Disorders: A Pilot Study. Bioengineering 2022, 9, 48. https://doi.org/10.3390/bioengineering9020048

Response 1: Thank you for your comments. We edited the limitations as suggested and included the reference that you mentioned.

Round 3

Reviewer 2 Report

I do not have any comments on the present form.